# Effect of Hyaluronic Acid and Kappa-Carrageenan on Milk Properties: Rheology, Protein Stability, Foaming, Water-Holding, and Emulsification Properties

**DOI:** 10.3390/foods12050913

**Published:** 2023-02-21

**Authors:** Suresh G. Sutariya, Prafulla Salunke

**Affiliations:** Dairy and Food Science Department, South Dakota State University, Brookings, SD 57007, USA

**Keywords:** hyaluronic acid, kappa-carrageenan, rheology, protein stability, water holding, oil emulsion, foaming

## Abstract

Hyaluronic acid (HA) is now widely known for its ability to bind water and impart texture. The combined effects of HA and kappa-carrageenan (KC) have not yet been investigated, though. In this study, we looked at the synergistic effects of HA and KC (concentrations of 0.1 and 0.25%, and ratios of 85:15, 70:30, and 50:50 for each concentration) on the rheological properties, heat stability, protein phase separation, water-holding capacity, emulsification properties, and foaming properties of skim milk. When HA and KC were combined in various ratios with a skim milk sample, this resulted in lesser protein phase separation and a higher water-holding capacity than when HA and KC were utilized separately. Similarly, for the sample with a 0.1% concentration, the combination of HA + KC blends demonstrated a synergistic impact with greater emulsifying activity and stability. The samples with a concentration of 0.25% did not exhibit this synergistic effect, and the emulsifying activity and stability were mostly due to the HA’s higher emulsifying activity and stability at 0.25% concentration. Similarly, for rheological (apparent viscosity, consistency coefficient K, and flow behavior index n) and foaming properties, the synergistic effect of the HA + KC blend was not readily apparent; rather, these values were mostly due to an increase in the amount of KC in the HA + KC blend ratios. When HC-control and KC-control samples were compared to various HA + KC mix ratios, there was no discernible difference in the heat stability. With the added benefits of protein stability (reduced phase separation), increased water-holding capacity, improved emulsification capabilities, and foaming abilities, the combination of HA + KC would be highly helpful in many texture-modifying applications.

## 1. Introduction

In Japan and South Korea, hyaluronic acid (HA) is recognized as a food additive and health food, respectively. HA is available as a dietary supplement in Belgium, Canada, Italy, and the United States [1]. In 2021, China approved HA as a food and beverage additive. In the US a proposal to give HA the GRAS designation, allowing for its usage in foods and beverages, is currently being reviewed by the FDA [2]. Given that HA is becoming more and more accepted in food applications, and its health advantages, learning more about its functional properties in dairy and food applications is very beneficial. In an aqueous environment, HA forms polymer spheres by occupying a high hydrodynamic volume because of the repulsive force between the negative charge of the carboxyl group and the intramolecular hydrogen bond [3]. This negative charge is due to the carboxylic acid group which helps HA molecules bind large amounts of water molecules. Because of its large water-binding ability, HA forms a highly viscous gel when mixed with an aqueous solution [4,5].

We are employing skim milk as a platform to investigate the functional advantages of HA in milk systems because of its capacity to build a viscoelastic network and increased water-binding capability in aqueous solutions. The knowledge from this milk system can also be applied to various dairy products in the future. Our latest research revealed that although HA made milk viscous, it had a negative impact on protein stability (phase separation) during storage [6]. KC is well known to stabilize the milk protein through its interaction with casein micelle when used in low concentrations [7]. Our hypothesis is that combining HA and KC may help address the issue of protein stability and enable the use of HA in dairy products to take advantage of its distinct rheological properties and health advantages. In addition to protein stability during shelf-life storage, protein stability during heat processing is crucial, and specific hydrocolloids are known to alter the heat stability of milk proteins during heat processing [8,9]. We conducted this investigation to further understand the impact of the HA + KC blend on milk heat stability because it was found in our earlier work that HC negatively affected the heat stability of milk protein [6]. Apart from rheological and textural benefits, hydrocolloids are also important in providing functional benefits such as foaming and emulsification properties. Our interest in thoroughly researching the synergistic role of the HA + KC blend in increasing the foaming and emulsification capabilities was sparked when preliminary experiments revealed that a combination of HA and KC created stable foams. The water-holding capacity and frequency sweep data were collected to understand the interaction of the HA + KC blend in the skim milk system and its effect on different functional properties.

The study was designed with the objective to determine the effect of different concentrations and ratios of HA and KC blends on various physicochemical properties of milk such as (i) protein phase separation during storage and heat stability during processing, (ii) viscosity profile and hydrocolloid interactions in a milk environment, (iii) foaming properties, (iv) water-holding capacity, and (v) emulsification properties. 

## 2. Materials and Methods

### 2.1. Design of Experiment

The study used a blend of HA and KC at two different concentrations (0.1% and 0.25%), along with five different HA:KC ratios for each concentration level. The HA:KC ratios used were 100:0 (HA-control), 85:15, 70:30, 50:50, and 0:100. (KC-control). Investigations were conducted on milk’s viscosity, protein stability during storage and processing, water-holding capacity, foaming potential, and emulsification qualities. Figure 1 graphically displays the experiment’s design.

### 2.2. Milk Sample Preparation

The sample of skim milk was bought from a nearby supermarket (Hy-Vee, Brookings, Washington, DC, USA). The preparation of samples and their analysis were performed as per the procedure described by Sutariya and Salunke [6]. Using Milkoscan model TM FT3 (Foss analytical A/S, Hillerød, Denmark), the contents of the skim milk samples’ fat, protein, lactose, and total solids were determined. The Kjeldahl standard method, as described by [10], was used to determine the total protein content, and the pH was measured using a pH meter (Hanna edgeblu, Smithfield, VA, USA). To preserve milk samples, a small quantity (0.02%) of sodium azide was added. The HA ~1500 kDa (Talsen Chemicals, New York, NY, USA) and KC powder (Cape crystal brands, Summit, NJ, USA) were dissolved in the milk to create samples with various concentrations and ratios of HA and KC as described in the experiment design (Figure 1). Pre-weighed amounts of HA and KC powder were mixed in cold milk (5 °C) at 25,000 rpm for 3 min using a high shear homogenizer POLYTRON^®^pt 2500 E (Kinematica AG, Malters, Switzerland). The samples were examined after mixing to ensure that the powders had been completely distributed and there were no discernible lumps. Following the mixing stage, the samples underwent a 20 s heat treatment at 80 °C before being quickly cooled in an ice-water bath (1 °C). After the heat treatment, these samples were hydrated in the refrigerator (5 °C) for 12 h. The final pH of these milk samples was adjusted to 6.7 using either 1 M HCl or 1 M NaOH while the milk was being stirred continuously. These samples were tested for gravimetric phase separation (protein stability), viscosity profile, frequency sweep, heat coagulation time (HCT), water-holding capacity, foaming capacity, foaming stability, emulsion activity, and emulsion stability analyses. 

### 2.3. Frequency Sweep 

A slightly modified version of the frequency sweep test described by Sutariya and Salunke [6] was used to better understand the time-dependent behavior and interactions of HA and KC polymers in the milk environment. The frequency sweep was carried out at a constant shear strain of 0.5% and over an angular frequency range of 16.7 to 89.5 rad/s. The sample temperature during the entire test was maintained at 5 °C. Values of the storage modulus (G’, Pa) and the loss modulus (G″, Pa) were plotted against angular frequency range values.

### 2.4. Protein Phase Separation by Gravimetric Settling Method during Storage 

A slightly modified version of the method published by Sutariya and Salunke [6] was used to assess the protein phase separation by gravimetric settling (protein stability). The samples were weighed (50 g), placed in graduated measuring cylinders, and left undisturbed for 48 h in a refrigerator maintained at 5 °C to allow gravimetric settling of proteins and phase separation due to the effect of HA + KC in the milk environment. Following 48 h, these samples were photographed to help see the protein phase separation. The 15 g sample was then carefully removed from the top layer and its total protein content was analyzed [10]. The percentage of protein phase separation was calculated using the equation below.
(1)$$\%\;\mathrm{Protein}\;\mathrm{phase}\;\mathrm{separation}\;=\left\{\frac{\left(\mathrm{Protein}\;\mathrm{in}\;\mathrm{control}\;\mathrm{sample}\;-\;\mathrm{Protein}\;\mathrm{in}\;\mathrm{top}\;15\;\mathrm{mL}\;\mathrm{layer}\;\right)}{\mathrm{Protein}\;\mathrm{in}\;\mathrm{control}\;\mathrm{sample}}\;\right\}\times 100\;$$

### 2.5. Heat Stability Measured by Heat Coagulation Time Test (HCT)

The HCT of the skim milk samples was determined using a method described by Sutariya and Patel [11] with slight modifications. The 5 g of sample was filled in 8 mL Wheaton glass tubes (D-17 mm × H-61 mm), airtight sealed, and clamped on the rocker stand. The rocker stand was submersed in an oil bath (Narang Scientific works PVT. LTD., New Delhi, India) maintained at 140 °C and put on rocker speed 3. The time needed to develop observable precipitation or coagulation at 140 °C was designated as the HCT.

### 2.6. Viscosity Profile Measurement

The method outlined by Sutariya and Huppertz [12] was slightly adjusted to measure the viscosity profile of these samples. For the viscosity profile measurement, a ramp liner shear rate profile range from 10 to 1000 s^−1^ and a testing temperature of 5 °C were used. The shear rate was increased by 20 s^−1^ over a liner time ramp duration from 2 to 10 s. The viscosity tests were performed using an MCR-92 rheometer (Anton Paar GmhH, Graz, Austria) and an analytical geometry of concentric cylinder (CC39, 38.690 mm) and cup (C-CC39, 42.010 mm), operated by Anton Paar RheoCompass 1.20 system. The viscosity profiles were assessed using a power law model as described by Sutariya and Salunke [6] in order to compare the samples’ viscous nature at a low shear rate and non-Newtonian viscosity behavior. 

### 2.7. Water-Holding Capacity

The milk samples were heated in a hot water bath to a temperature of 30 °C and maintained there for 15 min. The 10 mL of sample was filled in a graduated centrifuge tube. The samples were centrifuged at 30 °C for 15 min at 2000× *g* using sorvall ST plus series centrifuge (ThermoFisher Scientific, Karlsruhe, Germany). At the end of the centrifugation, the volume of sediment was recorded, and it was stated by the following Equation (2) [13].
(2)$$\mathrm{Water}\;\mathrm{holding}\;\mathrm{capacity}\;\%\;=\left(\frac{\mathrm{Volume}\;\mathrm{of}\;\mathrm{sediment}}{\mathrm{Initial}\;\mathrm{sample}\;\mathrm{volume}}\right)\times 100$$

### 2.8. Oil Emulsifying Activity and Emulsion Stability

The milk and sunflower oil (Hy-Vee, Brookings, USA) were first brought to a temperature of 30 °C in a hot water bath and kept for 15 min. Emulsion activity and emulsion stability were determined according to the method described by Mao and Hua [14] with some modifications. The 35 g of milk sample was mixed with 15 g of sunflower oil so that the 30% (*w*/*w*) oil-in-water emulsion could be tested [15]. To obtain the emulsion, the milk and oil samples were homogenized at a speed of 15,000 rpm for 2 min at 30 °C using a high shear homogenizer POLYTRON^®^pt 2500 E (Kinematica AG, Malters, Switzerland). The 10 mL of sample was filled in a graduated centrifuge tube. The samples were centrifuged at 30 °C for 15 min at 2000× *g* using sorvall ST plus series centrifuge (ThermoFisher Scientific, Karlsruhe, Germany). The volume of the emulsion was recorded, and it was stated by the following Equation (3).
(3)$$\mathrm{Oil}\;\mathrm{emulsion}\;\mathrm{activity}\;\%\;=\left(\frac{\mathrm{Volume}\;\mathrm{of}\;\mathrm{emulsion}\;\mathrm{layer}}{\mathrm{Initial}\;\mathrm{sample}\;\mathrm{volume}}\right)\times 100$$

Once the emulsion activity test was completed, the emulsion stability was determined by gently missing the aqueous and emulsion later in the same test tubes and re-centrifugation followed by heating at 80 °C for 30 min. The volume of the remaining emulsion was recorded, and it was stated by the following Equation (4).
(4)$$\mathrm{Oil}\;\mathrm{emulsion}\;\mathrm{stability}\;\%\;=\left(\frac{\mathrm{Volume}\;\mathrm{of}\;\mathrm{remaining}\;\mathrm{emulsion}\;\mathrm{layer}}{\mathrm{Initial}\;\mathrm{sample}\;\mathrm{volume}}\right)\times 100$$

### 2.9. Foaming Capacity and Foaming Stability

Foaming capacity and foaming stability were based on the method described by Mao and Hua [14] with slight modifications. The milk sample was first brought to a temperature of 30 °C in a hot water bath and kept for 15 min. The 100 mL milk samples were filled in a 250 mL graduated plastic beaker. The foaming was generated by mixing the sample in the same container at a speed of 15,000 rpm for 2 min at 30 °C using a POLYTRON^®^pt 2500 E homogenizer (Kinematica AG, Malters, Switzerland). Foaming capacity was determined by measuring the volume of foam instantly after 1 min of mixing. Then, it was stated by the following Equation (5). To study the foaming stability, the same samples were stored in a water bath maintained at 30 °C. Foaming stability was determined by measuring the reduction of the foam volume at intervals of 0.5, 1, 1.5, 2, 4, 5, 6, 18, and 24 h. Then, it was stated by the following Equation (6).

In addition to the foaming capacity and stability at 30 °C, the foaming capacity and stability at 65 °C for the 0.1% (HA + KC) was also determined to understand the foaming capacity and stability for the application of skim milk in coffee preparations.
(5)$$\mathrm{Foaming}\;\mathrm{capacity}\%\;=\left\{\frac{\left(\mathrm{Sample}\;\mathrm{volume}\;\mathrm{after}\;\mathrm{foaming}\;-\;\mathrm{Initial}\;\mathrm{sample}\;\mathrm{volume}\right)}{\mathrm{Initial}\;\mathrm{sample}\;\mathrm{volume}}\right\}\times 100$$
(6)$$\mathrm{Foaming}\;\mathrm{stability}\;\left(\mathrm{retention}\right)\%\;=\left\{1-\left[\frac{\left(\mathrm{Sample}\;\mathrm{volume}\;\mathrm{after}\;1\min\mathrm{offoaming}\;-\;\mathrm{Sample}\;\mathrm{volume}\;\mathrm{after}\;x\min\mathrm{offoaming}\right)}{\mathrm{Sample}\;\mathrm{volume}\;\mathrm{after}\;1\min\mathrm{offoaming}}\right]\right\}\times \;100$$
x = 0.5, 1, 1.5, 2, 4, 5, 6, 18, and 24 h.

### 2.10. Statistical Analysis

Each experiment was repeated three times. All the outcomes were examined using Minitab^®^ statistical software (version 0.3.1). One-way ANOVA with a two-sided confidence interval, 95% confidence level, and Tukey comparison technique was used to analyze the statistical differences between the samples with various dosages of HA treatments. Differences were deemed significant when *p*-values were less than 0.05 [6].

## 3. Results

### 3.1. Frequency Sweep

The frequency sweep is a useful technique for understanding the interpolymer interactions of HA + KC blends (at different concentrations and ratios) in the milk system. Frequency sweep results for 0.1% concentration and different ratios of HA + KC are displayed in Figure 2A,B. For the samples with 0.1%, HA + KC concentration, the higher G” values over G’ values of the HA-control, 85:15, and 70:30 ratio (HA:KC) indicated the dominating viscoelastic liquid behavior of these samples and hence the absence of gel network. The sample with a 50:50 ratio (HA:KC) showed a shift to viscoelastic solid behavior (G’ > G”) at a lower frequency range (<43.7 rad/s) and back to viscoelastic liquid behavior (G” > G’) at higher frequency (>43.7 rad/s). This viscoelastic solid behavior at the lower frequency indicated the presence of a weak inter-polymer network, which was mainly attributed by KC considering that the KC-control sample had a viscoelastic solid behavior (G’ > G”) through the complete frequency range (indicating the formation of stable inter-polymer network). Frequency sweep results for 0.25% concentration and different ratios of HA + KC are displayed in Figure 3A,B. Similar to 0.1% HA + KC concentration, the viscoelastic liquid behavior (G” > G’) for the HA-control and 85:15 samples were also observed for the 0.25% HA + KC concentration. The samples with 70:30 (HA:KC) and 50:50 (HA:KC) showed viscoelastic solid behavior (G’ > G”) through the entire frequency range, where again the viscoelastic solid behavior was mainly attributed to KC considering that the highest viscoelastic solid behavior was observed with KC-control sample. These results also indicated that the HA and KC polymers in blended samples did not show a synergistic effect in forming the stronger gel network as compared to the KC-control sample. 

### 3.2. Protein Phase Separation 

The gravimetric phases separation method was employed to examine the impact of HA + KC concentrations (0.1% and 0.25%) and various ratios on protein stability in the milk system during storage. The results (Figure 4) obtained from the protein phase separation study provided important information about the effect of these two hydrocolloids on the stability of the protein in milk during storage. For both concentrations (0.1% and 0.25%), HA-control samples showed the highest phase separation among all the samples followed by KC-control samples. The higher phase separation in the HA-control sample could have been due to the depletion flocculation phenomenon [6]. In the case of the KC-control sample, this may be related to the reduced KC–protein interaction at higher KC concentrations (>0.018%) [16]. For the samples with 0.1% concentration, the blend of HA + KC at all three ratios showed significantly (*p* < 0.05) lower phase separation as compared to both HA- and KC-control samples. Similarly, for the samples with 0.25% concentration, the blend of HA + KC at all 3 ratios also showed significantly (*p* < 0.05) lower phase separation as compared to both HA- and KC-control samples. Lower concentration of KC (<0.05%) is typically recommended to minimize the milk protein phase separation through weak gel network formation. The application of higher KC concentration can lead to phase separation [17]. Similarly, HA when used in a concentration > 0.05% leads to an increase in milk protein phase separation [6]. The increased milk protein phase separation at higher concentrations of HA-control and KC-control (0.1 and 0.25%) was in alignment with the finding of these studies. Interestingly, when HA and KC were used in combinations at higher concentrations (0.1 and 0.25%) it showed a significant (*p* < 0.05) reduction in the milk protein phase separations. The lowest phase separation at both concentrations in the samples with a 70:30 ratio indicated the best synergistic effect in reducing the phase separation as compared to HA and KC when used alone. Hence, combinations of HA and KC will allow its application at higher concentrations (>0.1%) to gain the functional benefits of these hydrocolloids without having a significant impact on milk protein phase separation. The frequency sweep results (Figure 2 and Figure 3) indicated that the significantly reduced phase separation of the blended HA + KC samples were independent of their viscoelastic liquid behavior (G” > G’, at ratios of 85:15 and 70:30 with 0.1% concentration, 85:15 ratio with 0.25% concentration) and viscoelastic solid behavior (G’ > G”, 50:50 ratio with 0.1% concentration, 70:30 and 50:50 ratio with 0.25% concentration), which indicated the possibility of protein stabilization through intertwined polymer network of HA and KC in the blended sample with the proteins in stabilizing them in the milk environment. 

### 3.3. Heat Coagulation Time Test (HCT)

The heat stability test (HCT at 140 °C) was carried out to understand the effect of HA and KC concentrations and ratios on the protein stability when subjected to the ultra-high temperature. The HCT results are displayed in Figure 5. For the samples with 0.1% concentration, the KC-control sample showed the highest HCT among all the samples. The increase in HCT with 50:50 (HC:KC) ratio at 0.1% concentration was mainly due to the contribution of KC. At a lower concentration level of 0.1%, the KC is known to interact with casein and thereby potentially increase the ζ potential which can improve heat stability through increased electrostatic repulsion [16,18,19]. In the case of samples with 0.25% (HA + KC) concentration, the results were somewhat opposite to the results we observed at 0.1% concentration. At a higher concentration level of 0.25%, the KC is known to have reduced interaction with casein micelle [16]; however, in this case, the negatively charged HA could possibly enhance the heat stability through increased electrostatic repulsion via the HA-casein complex [20]. Although all the samples showed lower heat stability (121 to 156 s) as compared to milk samples (395 s) [6] it was still above the commercial UHT processing requirements and hence should not be a concern for heat processing. The mechanism behind these decreases in HCT is not well understood. However, one of the potential factors could be the increase in calcium ions and protein and concentrations in the continuous serum phase as a result of increased water binding by HA and KC [6,21,22].

### 3.4. Flow Behavior Properties

The change in the viscosity behavior of milk samples as a function of different HA and KC concentrations and ratios is depicted in Figure 6A,B (0.1% concentration and 0.25% concentration). For a better understanding and numerical comparison of the shear thinning behavior and apparent viscosity at lower shear rates, flow behavior index *n* and the consistency coefficient log *K* (*Pa s^n^*) values were derived using a power law model (Table 1). The findings showed that log *K* values of the milk samples were significantly higher (*p* < 0.05) and *n* values were significantly lower (*p* < 0.05) as a function of HA and KC blend concentration (0.1 v/s 0.25%) for each HA:KC ratio (HA:KC ratio: 100:0, 85:15, 70:30, 50:50, 0:100). Moreover, for different ratios of HA:KC at each concentration level (0.1 and 0.25%), the log *K* values increased, and *n* values decreased as a function of proportions of KC in the ratio. Moreover, the highest log *K* values and lowest *n* values of the KC-control samples at both concentrations (0.1% and 0.25%) indicated that the amount of KC in different ratios (HA:KC) contributed to the increase in log *K* values and reduction in *n* values. The *n* values were highest for the HA-control samples at both concentrations (0.1 and 0.25%) levels. The significant (*p* < 0.05) reduction in *n* values was observed as a function of the increase in KC amount in the samples with different ratios of HA:KC (85:15, 70:30, 50:50, and KC-control). The frequency sweep results (Figure 2 and Figure 3) showed the absence of a gel network formation (G” > G’, viscoelastic liquid behavior) for HA-control samples and the presence of a gel network formation (G’ > G’’, viscoelastic solid behavior) for KC-control samples at both concentrations (0.1 and 0.25%), which can very well explain that for each HA:KC ratio (HA:KC ratio: 100:0, 85:15, 70:30, 50:50, 0:100) the increase in *K* values and reduction in *n* values was the function of KC’s ability to form gel network as the amount of KC increased in the HA + KC blended sample. Although, a combination of HA + KC showed a synergistic effect in reducing the protein phase separation; however, no synergistic effect was observed in viscosity profile results (Figure 6 and Table 1). 

### 3.5. Water-Holding Capacity

The water-holding capacity of the skim milk sample was influenced by the effect of HA + KA concentration as well as the ratios. The water-holding capacity results are displayed in Table 2. The water-holding capacity significantly (*p* < 0.05) increased as a function of concentration (0.1 v/s 0.25) at each ratio (HA:KC—100:0, 85:15, 70:30, 50:50), except for the KC-control sample. When comparing the HA-control with KC-control samples, the KC-control samples showed a higher water-holding capacity at both concentrations (Table 2). This higher water-holding capability of KC-control samples can be due to their gel formation capabilities at both concentrations, which was absent in the case of HA-control samples. For the samples with 0.1% concentration, the increase in water-holding capacity at different ratios was mainly contributed by the amount of KC, since the KC-control showed the higher water-holding capacity. However, for the samples with 0.25% concentration, the synergistic effect was demonstrated by the combination of HA + KC at all ratios (85:15, 70:30, and 50:50), which was evident by the significantly higher water-holding capacity (88 ± 1.7% at 85:15, 80 ± 0.0% at 70:30, and 62 ± 1.7% at 50:50) as compared to both HA-control (18 ± 1.4%) and KC-control (48 ± 3.8%). One possible explanation for the synergistic effect of HA + KC in retaining higher water could be that, while KC holds water through the gelling mechanism, the HA might be providing a synergistic effect by increasing the water binding (through hydrogen bonding with water molecules) in the serum phase trapped between the gel network. 

### 3.6. Emulsifying Activity and Stability

The emulsifying activity and stability results are displayed in Table 3. Emulsion activity and stability were significantly increased as a function of concentration (0.1 v/s 0.25%) at each ratio (HA:KC—100:0, 85:15, 70:30, 50:50, and 0:100). For 0.1% concentration, the blend of HA + KC samples showed significantly higher emulsion activity and stability as compared to both HA-control and KC-control, which indicated the synergistic effect of these ratios over individual HA and KC samples. For the sample with 0.25% concentration, both the emulsifying activity and stability were significantly higher for the HA-control samples including all three blended ratios (85:15, 70:30, and 50:50) as compared to the KC-control sample, which indicated that the higher emulsifying activity and stability in the blended samples were mainly contributed by the HA amounts. When comparing the stand-alone effect of HA v/s KC, for the 0.1% concentration, the HA-control sample showed significantly lower emulsion activity and stability as compared to KC-control which was reversed at 0.25% concentration. Based on these results, it appears that for the product formulation requiring lower concentration (≤0.1%), the use of a combination of HA + KC would yield better emulsion activity and stability. However, in the case of higher concentration (0.25%), HA alone would be able to provide the emulsion activity and stability similar to the combination of HA + KC. Overall, HA would be a great alternative or additional option in the space of hydrocolloids providing the dual benefit of texture and emulsification. The biopolymer such as HA and KC can act as emulsifiers through different modes of action such as forming a complex with the proteins covering the oil droplet surface and thereby creating strong repulsive forces between oil droplets [23,24] and their gel-forming capabilities slowing down the movement and separation ability of the oil droplets [25]. At 0.1% concentration, the emulsifying action of the KC could be mainly attributed to its complex formation with milk protein (interaction between negatively changed KC and positively charges region of κ-casein) located at the milk serum phase and oil interphase [26]. Along with the action of KC and κ-casein complex at the oil droplet surface, the viscosity increase in milk samples due to the action of KC would also contribute to KC’s ability to improve milk emulsification properties [25]. HA being a negatively charged biopolymer similar to KC, HA’s emulsification properties could also be through a similar mechanism of HA and κ-casein complex at oil droplet interphase and the viscosity increase in milk samples due to the action of HA. The lower emulsion activity and stability of the HA-control at 0.1% concentration compared to the KC-control sample could be due to the lower viscosity of the HA-control sample. The significantly higher emulsion activity and stability of the samples containing HA + KC blends with different ratios could be due to the combined synergistic effect of the increased viscosity of the serum phase (as a function of HA) trapped within a weak gel network formed by KC, which together will have a higher synergistic effect on slowing down the oil droplet movement and phase separation as compared to HA-control and KC-control samples. For the samples with 0.25% concentration, lower emulsification activity and stability of the KC-control sample as compared to the HA-control sample could be attributed to the reduced KC and κ-casein interactions [16], and hence the emulsification properties of the KC-control sample could be mainly attributed to gel network formation (G’ > G”, Figure 6B) and higher viscosity. The higher emulsification activity and stability of the HA-control and HA + KC blended sample could possibly be attributed to the interaction of HC and κ-casein complex (interaction between negatively charged HC and positively charged region of κ-casein) at the oil droplets—milk serum interphase in combination with higher viscosity. These synergistic benefits of higher emulsification of HA + KC blend would be of interest to the food manufacturers.

### 3.7. Foaming Capacity and Stability 

#### 3.7.1. Foaming Capacity and Stability of Skim Milk Sample at 65 °C

The effect of the milk treated with 0.1% HA + KC concentration and different ratios on the foaming properties was investigated targeting its application in coffee preparation. The most desirable method for coffee preparation requires heating the milk to 65–70 °C to partially denature the proteins to enhance the foam stability and also this temperature is a requirement for dispensing hot beverages [27,28]. For this purpose, the foaming capacity and stability of the skim milk samples (containing HA + KC) were studied at 65 °C. For this study, only 0.1% concentration was selected as the viscosity at 0.25% concentration was high for a coffee application. The foam capacity and foam stability data are displayed in Table 4. The foam capacity was lowest with the HC-control sample (98.3 ± 1.7%) and highest with the KC-control sample (170 ± 2.9%). For the sample containing different ratios of HA:KC, the foam capacity increased as a function of the increase in KC concentration in the ratio; however, they were still lower than the KC-control sample. This indicated the absence of a synergistic effect between HA and KC in a mixed blend to yield higher foaming compared to the KC-control sample. 

As for the retention of foam stability, the HA-control sample showed the lowest foam retention (79 ± 1.7% at 0.5 h mark and 59 ± 1.7% at 1 h mark) as compared to the KC-control sample (88 ± 2.9% at 0.5 h mark and 82 ± 2.9% at 1 h mark). The retention of foam (both at 0.5 h and 1 h mark) increased as a function of KC concentration in the blended samples. Overall, the foam retention of the 50:50 blend (HA:KC) was either equivalent (at 0.5 h mark) or better (at 1 h mark) as compared to the KC-control sample. We continued to monitor the foam retention for further time and all the foam subsided after 6 h for all the samples. Based on the results, HA does have a great potential for generating foam and better stability for application in coffee preparations. However, KC has a better performance which could be related to its ability to form a weak gel at a lower concentration.

#### 3.7.2. Foaming Capacity and Stability of Skim Milk Sample at 30 °C

The foaming capacity and stability of two different concentrations (0.1% and 0.25%) and different blend ratios of HA:KC (100:0, 85:15, 70:30, 50:50, 0:100) at 30 °C are displayed in Table 5 and Table 6. Overall, foaming capacity and stability were lower at 30 °C as compared to 65 °C, which could be a result of higher viscosity at a lower temperature. Moreover, the samples with 0.25% showed lower foaming capacity but higher foaming retention as compared to 0.1% concentration at 30 °C; again, this could be attributed to the higher viscosities of the sample with 0.25% concentration [29]. The HA-control samples at both concentrations showed lower foaming capacity as compared to the KC-control samples. For the samples with 0.1% concentration, the foam retention was 100% for all the samples at the 0.5 h mark, which decreased with the storage time up to 24 h. At the mark of 24 h storage, the HA-control and blended sample with 85:15 ratio (HA:KC) showed significantly lower foam retention as compared to the KC-control and blended samples with 70:30 and 50:50 ratios. For the samples with 0.25% concentration, the foam retention was 100% for all the samples up to the 1.5 h mark, which declined with the storage time up to 24 h. At the mark of 24 h storage, the HA-control and blended sample with 85:15 ratio (HA:KC) showed significantly (*p* < 0.05) lower foam retention as compared to the KC-control and blended samples with 70:30 and 50:50 ratios. The KC-control and blended samples with a 50:50 ratio showed the most stable (100%) foam retention over the entire testing period of 24 h. Based on the results it can be concluded that although HA does have a good foaming capacity and stability; however, it was still lower compared to KC, and improved foaming capacity and stability of the HA + KC blended samples (compared to HA-control) were mainly contributed by the KC component in the blend. Considering the gradual increase in the negative image of carrageenan, HA does have the potential of providing an alternate ingredient for foaming applications. 

In a skim milk environment during high-speed mixing, the milk proteins could diffuse from the serum phase and adsorb on the air-water interface to facilitate the foam formation. Adding negatively charged hydrocolloid (HA and KC) could enhance the flexibility of diffusion and distribution of proteins at the gas–water interface and facilitates the formation of an adsorption layer, which promotes capturing and formation of bubbles [24]. Hydrocolloids (HA and KC) being hydrophilic molecules does not directly participate in the foam formation; however, they could interact with milk protein molecules (interaction between the negatively charged region of hydrocolloid and positively charged region of milk protein) and thereby inducing the conformational changes to the protein molecule which are more favorable to support foam formation [24]. These protein conformation changes caused by KC–protein interaction might be favoring more foam formation as compared to the protein conformation changes caused by HA–protein interaction, which is reflected in the higher foam capacity of the KC-control as compared to HA-control. The progressive increase in foam capacity of the blended sample (HA + KC) as a function of the increase in KC amount could be mainly related to the contribution of KC’s protein conformation changes caused by KC–protein interaction favoring progressively increased foam formation. For foam stability, factors such as drainage, coalescence, and disproportionation play a role in thinning of the interfacial film, leading to the breakage and collapse of the foam [24]. From frequency sweep results (G’ > G”, for KC-control at both 0.1 and 0.25% concentration) the KC’s gel network forming ability would slow down the drainage of the serum phase around the foam and thereby contribute to the longer stability [29,30]. Moreover, in the blended samples, the amount of KC in each blend will proportionally contribute to longer foam stability. Similarly, KC-induced gel in the serum phase could also possibly play a role in minimizing the coalescence and disproportionation and thereby contribute to longer foam stability.

## 4. Conclusions

The combination of HA + KC blends at all three ratios showed a very good synergistic effect in significantly (*p* < 0.05) reducing the protein phase separation as well as increasing the water-holding capacity as compared to HA and KC when used alone. Similarly, the combination of HA + KC blends showed a synergistic effect in significantly (*p* < 0.05) higher emulsifying activity and stability for the sample with 0.1% concentration. For the samples with 0.25% concentration, the emulsifying activity and stability were mainly attributed to the higher emulsifying activity and stability of the HA. Heat stability had no noticeable impact when HC-control and KC-control samples were compared with different ratios of HA + KC blends. The rheological (viscosity, *K* values, and *n* values) and foaming properties were higher with the KC-control and lowest with the HA-control sample; the increase in these values was mainly related to an increase in the amount of KC in the HA + KC blend ratios. The combination of HA + KC would be very beneficial in different texture-modulating applications with the added benefit of protein stability (lower phase separation), higher water-holding capacity, better emulsification properties, and foaming abilities. It would be interesting to explore the synergistic effect of HA of different molecular weights in combination with other different types of hydrocolloids.

## Figures and Tables

**Figure 1 foods-12-00913-f001:**
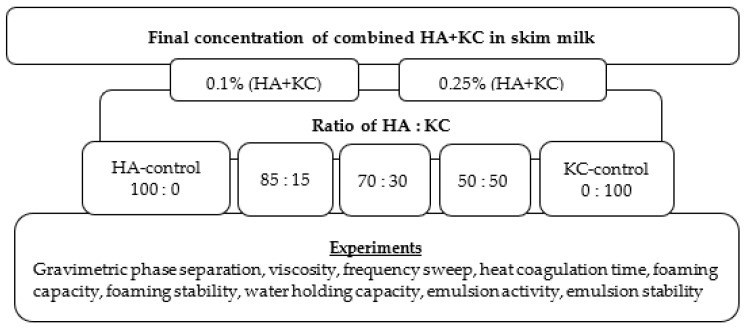
Experimental design.

**Figure 2 foods-12-00913-f002:**
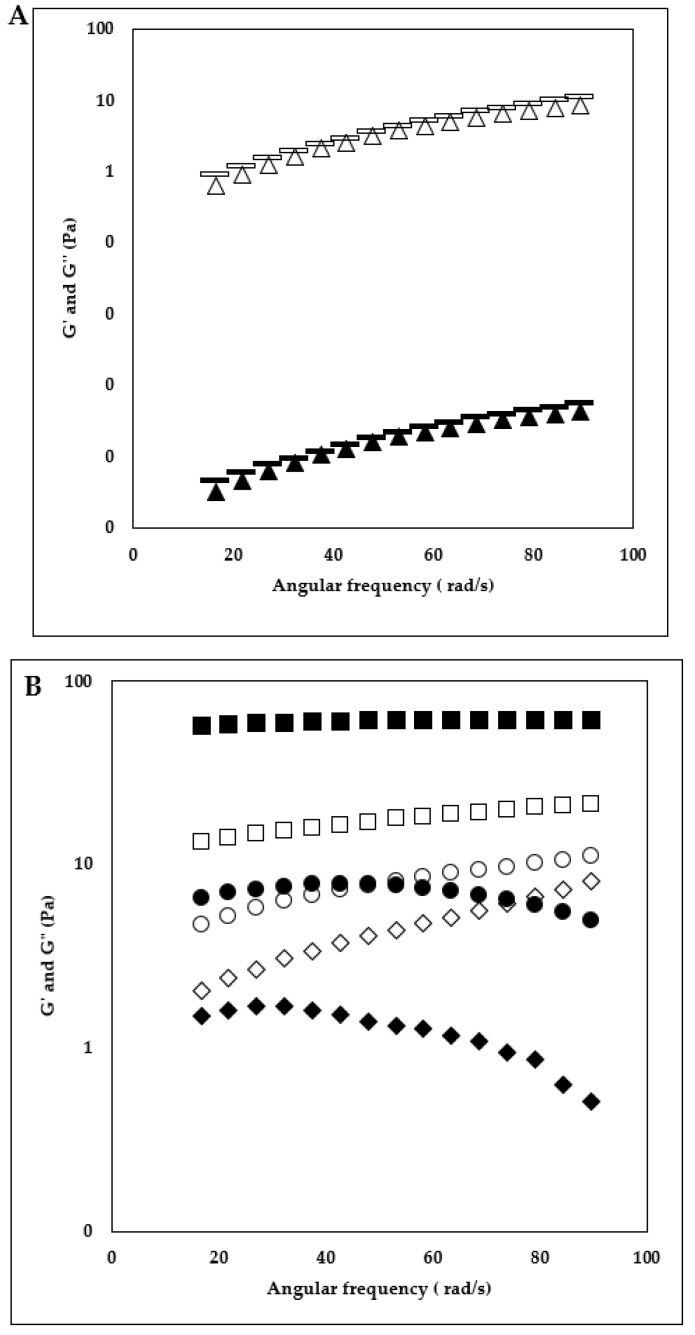
Frequency sweep test. G’—storage modulus and G”—loss modulus as a function skim milk sample treated with 0.1% (HA + KC) concentration and different ratios of HA:KC. (**A**) HA-control (100:0) G’ (▲), G” (∆), HA:KC (85:15) G’ (

) G” (

), (**B**) HA:KC (70:30) G’ (♦) G” (◊), HA:KC (50:50) G’ (●) G” (○), and KC-control (0:100) G’ (■) G” (□). To allow better visual clarity of the graph, the error bars (*n* = 3, for the standard error of the mean) are not shown.

**Figure 3 foods-12-00913-f003:**
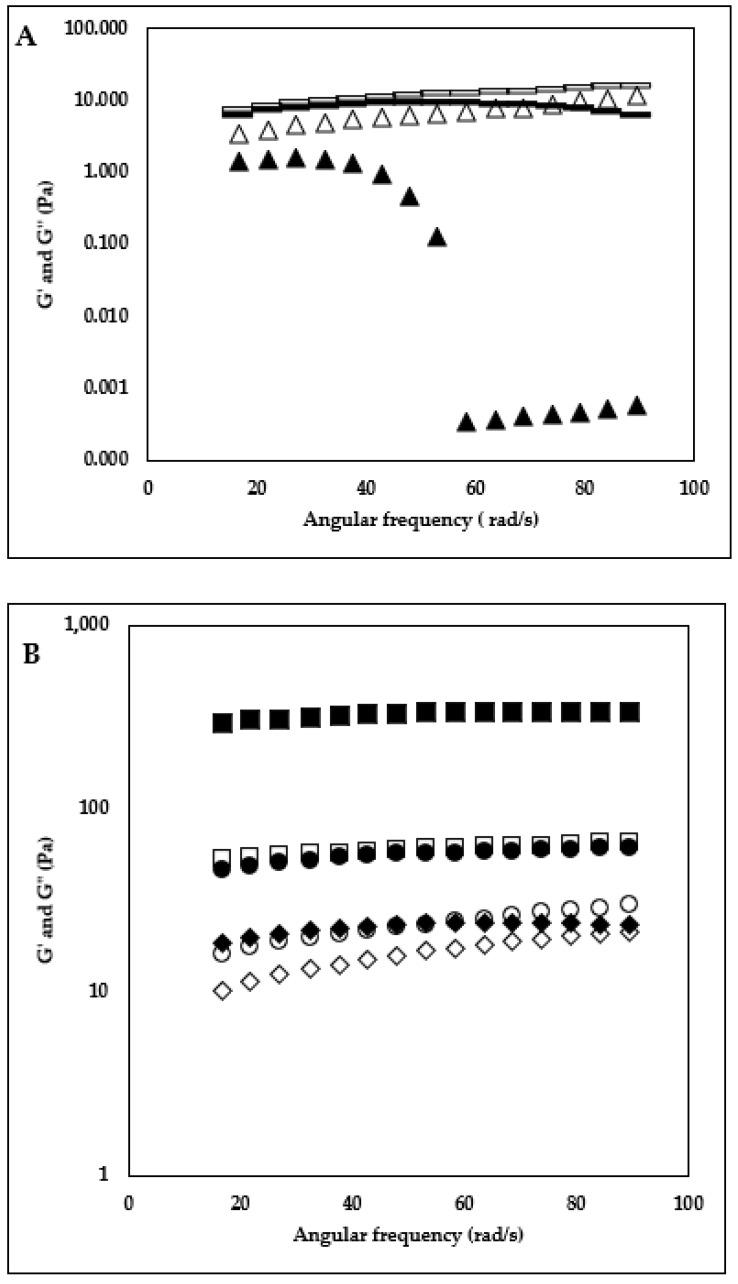
Frequency sweep test. Storage modulus (G’) and loss modulus (G”) as a function skim milk sample treated with 0.25% (HA + KC) concentration and different ratios of HA:KC. (**A**) HA-control (100:0) G’ (▲) G” (∆), HA:KC (85:15), G’ (

), G” (

), (**B**) HA:KC (70:30) G’ (♦) G” (◊), HA:KC (50:50) G’ (●) G” (○), and KC-control (0:100) G’ (■) G” (□). To allow better visual clarity of the graph, the error bars (*n* = 3, for the standard error of the mean) are not shown.

**Figure 4 foods-12-00913-f004:**
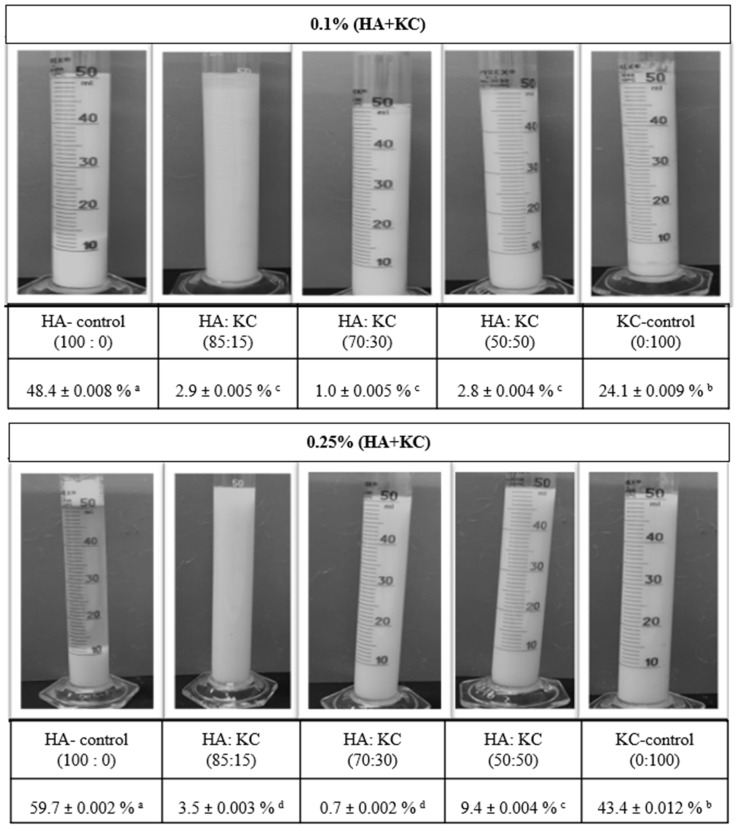
Protein phase separation of skim milk samples treated with different concentrations and ratios of HA + KC blend during storage (5 °C for 48 h). All values in this table are the mean (*n* = 3) ± the standard error of the mean. ^a–d^ completely different superscript letters in a table show significant differences (*p* < 0.05).

**Figure 5 foods-12-00913-f005:**
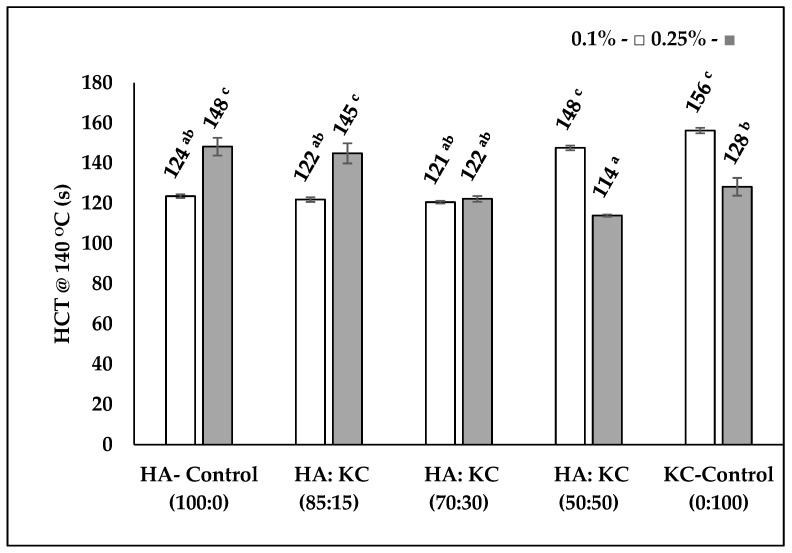
HCT at 140 °C of skim milk sample as a function of two different concentrations of 0.1% (□) and 0.25% (■), and different ratios of HA + KC: The error bars represent the standard deviation of the mean for each treatment’s triple analysis. Significant differences between (*p <* 0.05) HCT values on the column bar are shown by completely different superscript letters ^a–c^.

**Figure 6 foods-12-00913-f006:**
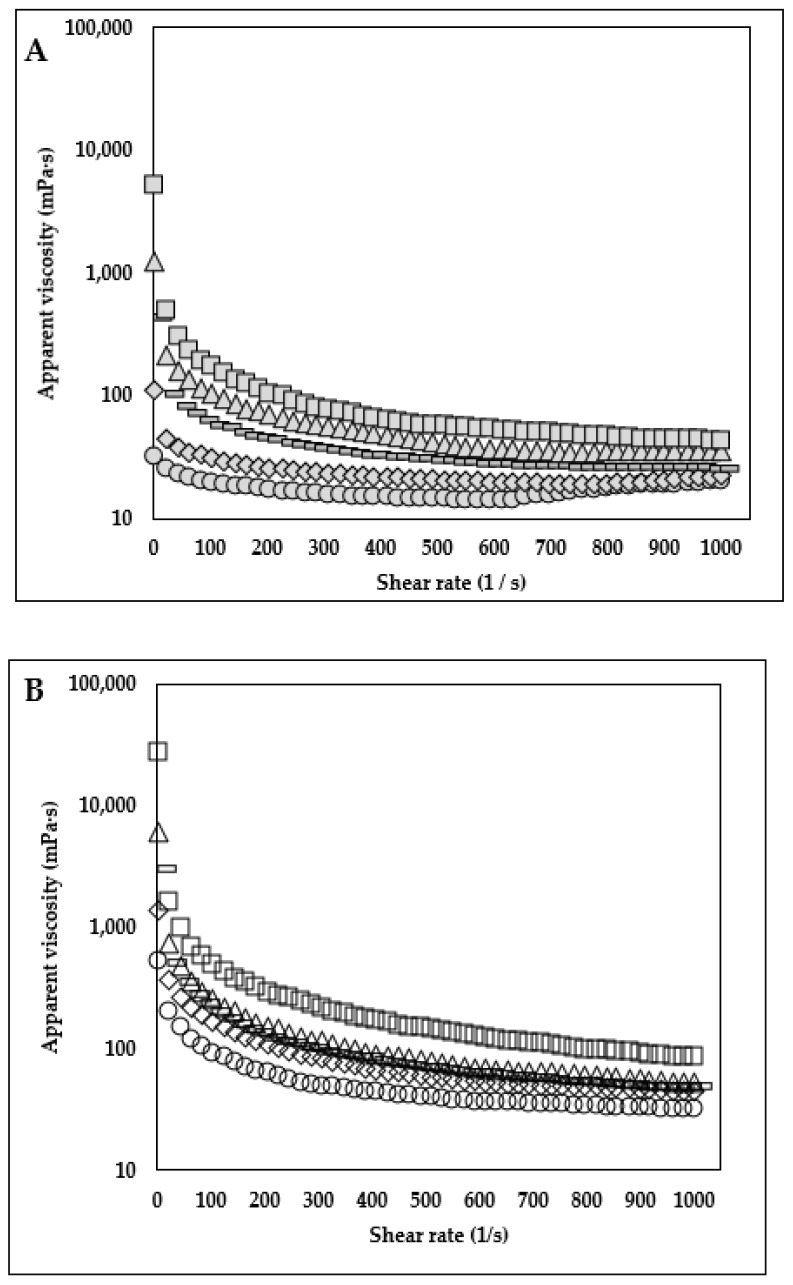
Viscosity profile of skim milk samples treated with different concentrations and ratios of HA:KC. (**A**) 0.1% HA + KC: HA-control (100:0)—(●), HA:KC (85:15)—(♦), HA:KC (70:30)—(

), HA:KC (50:50)—(▲), and KC-control—(■). (**B**) 0.25% HA + KC: HA-control (100:0)—(○), HA:KC (85:15)—(◊), HA:KC (70:30)—(

), HA:KC (50:50)—(∆), and KC-control—(□). To allow better visual clarity of the graph, the error bars (*n* = 3, for the standard error of the mean) are not shown.

**Table 1 foods-12-00913-t001:** Power law-derived consistency coefficient (*K*) and flow behavior index (*n*) of skim milk samples treated with different concentrations and ratios of HA + KC blend.

HA:KC (Ratio)	log *K*(*Pa s^n^*)		*n* (-)	
	0.1% (HA + KC)	0.25% (HA + KC)	0.1% (HA + KC)	0.25% (HA + KC)
HA-control (100:0)	0.029 ± 0.00 ^a,A^	0.69 ± 0.07 ^a,B^	0.91 ± 0.00 ^a,C^	0.55 ± 0.02 ^a,D^
HA:KC (85:15)	0.093 ± 0.00 ^a,A^	1.81 ± 0.21 ^a,B^	0.77 ± 0.00 ^b,C^	0.46 ± 0.02 ^b,D^
HA:KC (70:30)	0.38 ± 0.01 ^a,A^	3.37 ± 0.05 ^ab,B^	0.60 ± 0.00 ^c,C^	0.38 ± 0.00 ^c,D^
HA:KC (50:50)	1.13 ± 0.05 ^b,A^	6.11 ± 0.03 ^b,B^	0.48 ± 0.01 ^d,C^	0.31 ± 0.00 ^d,D^
KC-control (0:100)	4.01 ± 0.23 ^c,A^	22.0 ± 1.64 ^c,B^	0.33 ± 0.01 ^e,C^	0.18 ± 0.00 ^e,D^

HA = Hyaluronic acid; KC = kappa-carrageenan for all figures and tables. All values in this table are the mean of triplicate analyses ± the standard error of the mean. ^a–e^ Completely different superscript letters in the column show a significant difference (*p * < 0.05) between different ratios of HA and KC. Significant differences (*p* < 0.05) in log *K* values between 0.1% and 0.25% concentrations of HA and KC are shown by different superscript letters ^A,B^ in a row (for log *K only*). Significant differences (*p* < 0.05) in *n* values between 0.1% and 0.25% concentrations of HA and KC are shown by different superscript letters ^C,D^ in a row (for *n only*).

**Table 2 foods-12-00913-t002:** Water-holding capacity of skim milk samples treated with different concentrations and ratios of HA + KC blend.

% Water-Holding Capacity					
Concentration (HA + KC)	HA-Control (100:0)	HA:KC (85:15)	HA:KC (70:30)	HA:KC (50:50)	KC-Control (0:100)
0.1%	10 ± 0 ^a,A^	25 ± 0 ^b,A^	42 ± 1.7 ^c,A^	52 ± 1.7 ^d,A^	47 ± 1.7 ^cd,A^
0.25%	18 ± 1.4 ^a,B^	88 ± 1.7 ^b,B^	80 ± 0 ^b,B^	62 ± 1.7 ^c,B^	48 ± 3.8 ^d,A^

All values in this table are the mean of triplicate analyses ± the standard error of the mean. Significant differences (*p* < 0.05) for the row values are shown by superscript letters ^a–d^ and for the column values are shown by superscript letters ^A,B^.

**Table 3 foods-12-00913-t003:** Emulsifying properties (emulsifying activity and stability) of skim milk sample treated with different concentrations (0.1 and 0.25% HA + KC) and different ratios of HA + KC blend.

% Emulsifying Properties						
Concentration (HA + KC)	Emulsion Test Type	HA-Control (100:0)	HA:KC (85:15)	HA:KC (70:30)	HA:KC (50:50)	KC-Control (0:100)
0.10%	% Emulsion activity	55 ± 1.4 ^a,A^	72.5 ± 1.4 ^b,c,A^	79.2 ± 0.8 ^c,A^	76.7 ± 1.7 ^c,A^	66.7 ± 1.7 ^b,A^
	% Emulsion stability	50 ± 2.9 ^a,A^	65 ± 0 ^b,B^	75 ± 0 ^c,B^	79.2 ± 2.2 ^c,A^	60 ± 2.9 ^b,A,B^
0.25%	% Emulsion activity	95 ± 0 ^a,B^	96.7 ± 0 ^a,C^	95 ± 0 ^a,C^	90 ± 0 ^b,B^	75 ± 0.8 ^c,B,C^
	% Emulsion stability	87 ± 1.7 ^a,C^	86.7 ± 1.7 ^a,D^	90.8 ± 0.8 ^a,D^	93.3 ± 1.7 ^a,B^	76.7 ± 1.7 ^b,C^

All values in this table are the mean of triplicate analyses ± the standard error of the mean. Significant differences (*p* < 0.05) for the row values are shown by superscript letters ^a–c^ and for the column values are shown by superscript letters ^A–D^.

**Table 4 foods-12-00913-t004:** Foaming capacity and stability of the skim milk sample at 65 °C treated with 0.1% HA + KC concentration and different ratios of HA + KC blend.

	% Foam Capacity	% Foam Stability (Retention)			
HA:KC (Ratio)	0 h	0.5 h	1 h	6 h	12 h
HA-control (100:0)	98.3 ± 1.7 ^A^	79 ± 1.7 ^A,a^	59 ± 1.7 ^A,b^	54 ± 1.7 ^A^	0 ± 0 ^A,c^
HA:KC (85:15)	98.3 ± 1.7 ^A^	79 ± 1.7 ^A,a^	79 ± 1.7 ^B,a^	74 ± 1.7 ^B^	0 ± 0 ^A,c^
HA:KC (70:30)	118.3 ± 1.7 ^B^	83 ± 1.7 ^B,a^	75 ± 1.7 ^C,b^	66 ± 1.7 ^B^	0 ± 0 ^A,c^
HA:KC (50:50)	146.7 ± 1.7 ^C^	87 ± 1.7 ^C,a^	87 ± 1.7 ^D,a^	52 ± 1.7 ^B^	0 ± 0 ^A,c^
KC-control (0:100)	170 ± 2.9 ^D^	88 ± 2.9 ^D,a^	82 ± 2.9 ^E,a^	41 ± 2.9 ^B^	0 ± 0 ^A,c^

All values in this table are the mean of triplicate analyses ± the standard error of the mean. Significant differences (*p* < 0.05) for the row values are shown by superscript letters ^a–c^ and for the column values are shown by superscript letters ^A–E.^

**Table 5 foods-12-00913-t005:** Foaming capacity and stability of skim milk sample at 30 °C treated with 0.1% HA + KC concentration and different ratios of HA + KC blend.

	% Foam Capacity	% Foaming Stability (Retention)								
HA:KC (Ratio)	0 h	0.5 h	1 h	1.5 h	2 h	4 h	5 h	6 h	18 h	24 h
HA-control (100:0)	68 ± 1.7 ^a^	100 ± 1.7 ^a^	85 ± 1.7 ^a^	85 ± 1.7 ^a^	85 ± 1.7 ^a^	85 ± 1.7 ^a^	85 ± 1.7 ^a^	85 ± 1.7 ^a^	41 ± 1.7 ^a^	36 ± 1.7 ^a^
HA:KC (85:15)	78 ± 1.7 ^b^	100 ± 1.7 ^b^	87 ± 1.7 ^b^	87 ± 1.7 ^b^	81 ± 1.7 ^ab^	81 ± 1.7 ^ab^	81 ± 1.7 ^ab^	74 ± 1.7 ^a^	49 ± 1.7 ^a^	49 ± 1.7 ^a^
HA:KC (70:30)	80 ± 0 ^b^	100 ± 0 ^b^	87 ± 0 ^b^	87 ± 0 ^b^	87 ± 0 ^b^	87 ± 0 ^b^	87 ± 0 ^b^	87 ± 0 ^b^	87 ± 0 ^b^	87 ± 0 ^b^
HA:KC (50:50)	92 ± 1.7 ^c^	100 ± 1.7 ^c^	89 ± 1.7 ^c^	89 ± 1.7 ^c^	89 ± 1.7 ^c^	89 ± 1.7 ^c^	89 ± 1.7 ^c^	89 ± 1.7 ^c^	89 ± 1.7 ^c^	89 ± 1.7 ^c^
KC-control (0:100)	92 ± 1.7 ^c^	100 ± 1.7 ^c^	89 ± 1.7 ^c^	89 ± 1.7 ^c^	89 ± 1.7 ^c^	89 ± 1.7 ^c^	89 ± 1.7 ^c^	89 ± 1.7 ^c^	89 ± 1.7 ^c^	89 ± 1.7 ^c^

All values in this table are the mean of triplicate analyses ± the standard error of the mean. Significant differences (*p* < 0.05) for the column values are shown by superscript letters ^a–c^.

**Table 6 foods-12-00913-t006:** Foaming capacity and stability of skim milk sample at 30 °C treated with 0.25% HA + KC concentration and different ratios of HA + KC blend.

	% Foam Capacity	% Foam Stability								
HA:KC (Ratio)	0 h	0.5 h	1 h	1.5 h	2 h	4 h	5 h	6 h	18 h	24 h
HA-control (100:0)	60 ± 0 ^a^	100 ± 0 ^a^	100 ± 0 ^a^	100 ± 0 ^a^	100 ± 0 ^a^	80 ± 3.3 ^a^	75 ± 1.7 ^a^	63 ± 6.0 ^a^	33 ± 0 ^a^	33 ± 0 ^a^
HA:KC (85:15)	60 ± 0 ^a^	100 ± 0 ^a^	100 ± 0 ^a^	100 ± 0 ^a^	100 ± 0 ^a^	97 ± 1.7 ^b^	92 ± 0 ^c^	92 ± 0 ^b^	72 ± 1.7 ^b^	50 ± 0 ^b^
HA:KC (70:30)	57 ± 1.7 ^a^	100 ± 1.7 ^a^	100 ± 0 ^a^	100 ± 0 ^a^	95 ± 0 ^a^	88 ± 0 ^ab^	88 ± 0 ^b^	88 ± 0 ^b^	88 ± 0 ^c^	88 ± 0 ^c^
HA:KC (50:50)	72 ± 1.7 ^b^	100 ± 0 ^b^	100 ± 0 ^b^	100 ± 0 ^b^	100 ± 0 ^b^	100 ± 0 ^c^	100 ± 0 ^d^	100 ± 0 ^c^	100 ± 0 ^d^	100 ± 0 ^d^
KC-control (0:100)	82 ± 1.7 ^c^	100 ± 0 ^c^	100 ± 0 ^c^	100 ± 0 ^c^	100 ± 0 ^c^	100 ± 0 ^d^	100 ± 0 ^e^	100 ± 0 ^d^	100 ± 0 ^e^	100 ± 0 ^e^

All values in this table are the mean of triplicate analyses ± the standard error of the mean. Significant differences (*p* < 0.05) for the column values are shown by superscript letters ^a–e^.

## Data Availability

The data presented in this study are available on request from the corresponding author. The data are not publicly available due to privacy need.

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
