# Peer review of "Effect of Hyaluronic Acid and Kappa-Carrageenan on Milk Properties: Rheology, Protein Stability, Foaming, Water-Holding, and Emulsification Properties"

_foods, 2023, doi:10.3390/foods12050913_

Round 1

Reviewer 1 Report

The work deals with the study of different physicochemical properties of skim  milk affected by the addition of different amounts of Hyaluronic acid and kappa-carrageenan.

A similar study was published by authors in Food Hydrocolloids but regarding only the effect of HA. Authors conclude that similar studies using mixtures with other hydrocolloids should be carried out.

From the analytical point of view the paper is OK.

However I have the next observations:

The interest of the work should be more emphasized in the introduction: why the skim milk matrix was chosen, why HA deserves to be applied as a technological ingredient beyond its  potential functional properties.

The second paragraph of the introduction should be revised. It is focused in the aim of the work (which it is not a good idea because the aim of the work is in the next paragraph) with some repeated sentences.

The third paragraph  seems to be too similar to that of the previous work in food hydrocolloids. It should be rewritten.

M&M sections 2.1 and 2.2 should be revised. Figure 1 is not necessary. A section with the origin of every material should be included. The amounts of HA and KC added to the skim milk should be included (maybe in a table) to clarify the formulations of the different analyzed samples (it seems that there is no really a previous blend of the two hydrocolloids, the preparation of the formulations was through the addition of each ingredient to the milk).

Table headers should be revised including which information is in A and B.

Discussion. Maybe it could be enriched comparing the results with those obtained for other hydrocolloids used in other works with the same function.

Author Response

Dear respected Editor and reviewers,

We greatly appreciate your time, effort, expertise, and skills in helping us to strengthen our manuscript. We have carefully reviewed your comments and suggestion and provided our action/explanation highlighted below in blue text in the attachment.    

Reviewer 2 Report

Review on manuscript: foods-2152488

Effect of hyaluronic acid and kappa-carrageenan on milk properties: Rheology, protein stability, foaming, water holding, and emulsification properties

by  Suresh G. Sutariya, Prafulla Salunke

submitted to Foods

In the manuscript submitted for comments, the authors studied the effect of hyaluronic acid and kappa-carrageenan on selected properties of milk. 

 Detailed recommendation

line 13 – instead of “viscosity profile, frequency sweep” better will be: rheological properties,

line 21 – should be: rheological properties

line 22 – should be: apparent viscosity, symbols K and n should be explained,

equations 1 - 6 – equations should be presented in equation format, not as text,

line 165 – sweep frequency test does not analyze the time-dependent behavior of the system, only viscoelastic properties,

line 168 – have the measurements been made in the range of linear viscoelasticity?

line 170 – should be: angular frequency,

Results – should be presented in accordance with the methodology, authors should limit the repetition of numerical values,

Figure’s captions – should be placed under the figures,

Figure 2A – units are missing on the y-axis,

Figure 3 – what was the reason for such a sharp drop in the value of G' for the control sample?

line 448 – beet will be: flow properties,

line 453 – should be:  Pa sn

Table 1 – does the model used accurately describe the experimental data? what were the R2 values?

Author Response

Dear respected Editor and reviewers,

We greatly appreciate your time, effort, expertise, and skills in helping us to strengthen our manuscript. We have carefully reviewed your comments and suggestion and provided our action/explanation highlighted in the attachment in blue text.   
